# Target-free ligand scoring via one-shot learning

## Abstract

Scoring ligands in a library based on their structural similarity to a known hit compound is widely used in drug discovery following high-throughput screening. However, such "similarity search" relies on the assumption that structurally similar compounds have similar activities, and will therefore only retrieve ligands with hit-like affinity, requiring resource-intensive tweaking by medicinal chemists to reach a more active lead compound. We propose a novel approach, *One-Shot Ligand Scoring* (OSLS), that is much more capable of directly retrieving lead-like compounds from a library using a novel one-shot learning technique. For this new task, we design a Siamese-inspired neural architecture using two Transformer encoders without tied weights, a novel positional encoding-like mechanism, and a final prediction head. OSLS is able to score ligands by activity against a target without any target-specific knowledge beyond a single known activity value, a cost-effective approach to ligand-based or phenotypic drug discovery. We show that OSLS surpasses traditional similarity search as well as modern deep learning baselines on a simulated ligand retrieval task. Furthermore, we demonstrate the applicability of our approach on various drug discovery tasks that also involve ligand scoring, including drug repositioning, precision patient-level drug efficacy prediction, and even molecular generative modeling.

## 1 Introduction

Contemporary drug discovery is a costly and time-consuming process requiring billions of dollars per new approved drug. A significant portion of the total development cost is incurred in preclinical stages, where medicinal chemists identify one or more "hit" compounds from high-throughput screens that have activity against the target of interest and retrieve structural analogs to these hits from large molecular catalogs for further exploration and development, and eventually produce a lead compound after much optimization (de Souza Neto et al., 2020; Hughes et al., 2011). Currently, the concept of chemical similarity is critical to the retrieval of these structural analogs through pairwise similarity scoring between the hit compound and each compound in a library. Commonly used similarity metrics include the Tanimoto similarity computed between binary fingerprints of two molecules (Bajusz et al., 2015), as well other more specialized metrics and molecular featurizations (Cereto-Massagué et al., 2015; Nikolova & Jaworska, 2003; Maziarka et al., 2020; Jaeger et al., 2018; Coupry & Pogány, 2022; Gandini et al., 2022).

Retrieving structural analogs from a chemical library with such similarity scoring tends to yield compounds whose activity is similar to that of the initial hit, based on the concept that chemically similar compounds have similar activity (Johnson & Maggiora, 1990). However, hit compounds from experimental screens typically have low activities, e.g. binding affinities of $1 - 10$ $\mu$M, compared to the typical $1 - 10$ nM goal of preclinical drug development, a thousand-fold difference (Freire, 2015; Hughes et al., 2011). Thus, retrieving library compounds based purely on their similarity to an early-stage hit compound is not an optimal strategy, as this is expected to yield compounds with hit-like activity instead of the desired highly active compounds. Compound scoring via similarity is common for other related problems in drug discovery, such as drug repositioning (Jarada et al., 2020) and lead optimization Hughes et al. (2011), which also suffer from the problem of measuring similarity to a weakly active compound.

Here, we propose *One-Shot Ligand Scoring* (`OSLS`), an alternative to chemical similarity that predicts the activity of an experimentally uncharacterized query compound (e.g. a compound drawn from a chemical library) to an unseen target based on a single context compound and its experimentally known activity to that target (e.g. a hit from an experimental screen). Like standard chemical similarity-based scoring, `OSLS` shares the advantage of needing no information about a target protein (Zheng et al., 2013; Vijayan et al., 2021)). However, `OSLS` is distinct from standard measures of chemical similarity, because, by using a one-shot learning paradigm, it can assign the highest scores to the most active compounds, instead of those most similar to a weakly active hit.

More particularly, we

- introduce the novel formulation of ligand scoring as a one-shot regression problem, and argue for its utility over traditional similarity-based scoring
- design a novel architecture, `OSLS`, which addresses this problem by using a Siamese-inspired neural architecture to extract target information from the known activity of a context compound and use it to directly predict the activity of a query compound
- show that `OSLS` outperforms both similarity-based as well as modern deep learning scoring techniques in settings relevant to compound retrieval and the related tasks of drug repositioning, patient-level drug efficacy prediction, and generative modeling.

## 2 RELATED WORK

In this work, we focus on cases where information about a targeted protein (e.g. amino acid sequence or 3D structure) is not used for compound scoring — so-called "ligand-based" or "phenotypic" drug discovery (Sharma et al., 2021; Swinney & Anthony, 2011; Zheng et al., 2013). In this setting, drug discovery begins from one or a few existing compounds with some known activity, often obtained from screening or known natural ligands. In this case, scoring of additional compounds is currently done using either chemical similarity or N-shot learning approaches.

**Chemical similarity.** Chemical similarity is commonly used when compounds with a desired activity are known, and involves computing the pairwise similarity between the known actives and each compound to be scored. When using a highly active known compound, similarity acts as a surrogate measure of activity (Johnson & Maggiora, 1990), although this approximation fails as the known compound becomes less active. Computing similarity is commonly done with binary fingerprints (e.g. circular fingerprints, Rogers & Hahn (2010)), although this approach will often undesirably miss compounds with similar activity but different chemical scaffolds. For this reason, many other chemical features have been suggested for representing molecules, including simple molecular weight as well as more complex representations that capture information about molecular topology and 3D shape/charge (Khan et al., 2016; Li et al., 2012; Kohlbacher et al., 2021; Kearnes & Pande, 2016).

However, such approaches are, arguably, based more on chemical intuition than data, and choosing which of hundreds of molecular descriptors to use adds another level of uncertainty. For this reason, machine learning-based techniques have been proposed that derive chemical similarity in a data-driven fashion and thus offer promise to improve quality of similarity measurement. In particular, much work has been dedicated to learning molecular featurization in an unsupervised fashion, which can later be used for downstream tasks such as similarity (Jaeger et al., 2018; Huang et al., 2021; Li & Jiang, 2021; Morris et al., 2020). Due to their unsupervised nature, however, similarity measurements between machine learning-derived embeddings are not necessarily meaningful for activity, as structurally dissimilar molecules may have similar activities, and vice-versa. Because of this, more direct N-shot learning approaches (Schimunek et al., 2021; Altae-Tran et al., 2017; Stanley et al., 2021; Lee et al., 2022) have been proposed to leverage vast amounts of existing activity data toward the scoring of new ligands against novel targets.

**N-shot learning.** N-shot learning techniques directly use existing compounds (the "context", also called the "support set") to predict the activity of unknown compounds (the "query set") without relying on similarity as an imperfect surrogate of activity. The application of Siamese networks to one-shot learning, a form of N-shot learning involving a single context example, was first introduced in computer vision (Koch et al., 2015), and existing Siamese-based techniques, such as those

that act on sentences (Mueller & Thyagarajan, 2016), may be applied to string-based molecular representations. Schimunek et al. (2021) extend this technique to multiple context compounds for few-shot learning, also a form of N-shot learning, using a Siamese network to encode both the query and context molecules followed by a similarity function to perform classification. Altae-Tran et al. (2017) introduce an LSTM-based architecture to iteratively update compound embeddings, which are then used to make binary activity predictions. Li et al. (2019) classify two graphs, which may be molecular graphs, as similar or dissimilar through a cross-graph attention mechanism. Finally, Stanley et al. (2021) introduce a dataset for the few-shot classification task in drug discovery, and run multiple baselines on the proposed set.

All these techniques, however, cannot predict *how* active a compound is, instead making only binary predictions of activity at whatever threshold of true activity is chosen. This is especially relevant when known actives are only of mild potency so the "active" threshold is low, e.g. hits following a screen. In this case, binary methods cannot distinguish between mildly and highly active compounds. Lee et al. (2022) propose to predict continuous binding affinity in a few-shot fashion using attentive neural processes (ANPs), but their method requires multiple support examples and is not applicable to our one-shot setting. Additionally, the support set is randomly drawn over the entire activity range of known compounds, including highly active compounds, casting doubt on the model's utility in the case where one knows only a mildly active compound.

## 3 METHODOLOGY

We formulate the problem of compound scoring given a known active as a one-shot regression problem. We introduce `OSLS`, an architecture using two Siamese-like encoding networks with untied weights, a positional encoding-like mechanism to represent activity values, and a feedforward prediction head. The architecture is summarized in Figure 1.

**Problem statement.** We seek to score compounds via one-shot learning, meaning predicting, i.e. scoring, the activity of a "query" compound against an unseen target given only a single "context" compound and its activity against that target. Let $m$ represent a molecule and $\pi_P(m) \in \mathbb{R}$ its experimentally determined activity against a target $P$ (e.g. binding affinity to a protein, or phenotypic inhibitory concentration). Formally, we aim to predict the activity of a query molecule $m_q$ to $P$, $\pi_P(m_q)$, given a context molecule $m_c$ and its activity against $P$, $\pi_P(m_c)$. That is, we wish to learn a function $f(m_q, m_c, \pi_P(m_c)) \in \mathbb{R}$ to approximate the true activity of the query molecule, $\pi_P(m_q)$.

**Architecture.** We employ two separate encoders, a context and a query encoder, that feed into an activity prediction head. The context encoder, $f_c : (m_c, \pi_P(m_c)) \mapsto x_c$, learns to encode information about the target into a real-valued vector $x_c$ using the context molecule and its known activity against the target. The query encoder, $f_q : m_q \mapsto x_q$, learns to encode the query molecule into a representation $x_q$, that is useful for predicting its activity. As they are not designed to encode the same information, $f_c$ and $f_q$ do not share weights, unlike previously proposed Siamese networks (Mueller & Thyagarajan, 2016; Morris et al., 2020). The predictor network, $g : x_c \oplus x_q \mapsto \hat{\pi}_P(m_q)$, where $\oplus$ denotes vector concatenation, uses these two encodings to predict the activity of the query molecule, $\hat{\pi}_P(m_q) \in \mathbb{R}$.

We represent molecules as sequences of atom-level tokens of canonical SMILES strings (see Table 3 for comparison across molecular representations). Both $f_c$ and $f_q$ are Transformer encoders (Vaswani et al., 2017), which have shown recent success as a way to represent molecules Wang et al. (2019); Li & Jiang (2021). Following Devlin et al. (2018), we also prepend a "classification token" at the first position of each sequence, and take the hidden state of the Transformer at that position as $x_c$ and $x_q$ after projection through a linear layer.

To encode scalar activity information with the Transformer, we propose a novel, learned linear projection of the scalar that is summed with token embeddings before being fed to the multi-head attention of the Transformer encoder. This projection has the same dimensionality as the embeddings, and is identical across each element of the input sequence. This simple approach to including scalar activity information is similar to positional encoding, which can effectively pass contextual information to the Transformer (Vaswani et al., 2017).

**Training.** The training dataset, $\mathcal{D}$, consists of compounds and their affinities to multiple protein targets, $\mathcal{P}$. Let $\mathcal{M}_P$ represent all compound with activity data against a given target $P \in \mathcal{P}$, then we define the training set as

$$\mathcal{D} = \bigcup_{P \in \mathcal{P}} \{(m, \pi_P(m)) \mid m \in \mathcal{M}_P\}$$

where $\bigcup$ represents repeated union operations. The testing set is constructed in a similar way to the training set, but with a separate set of targets, a so-called "target split" (Feng et al., 2018).

We train the model in an end-to-end fashion with Mean Squared Error (MSE) loss between the predicted and true activity of $m_q$, i.e. $\text{MSE}(\pi_P(m_q), \hat{\pi}_P(m_q))$. Each training batch consists of a set of query molecules randomly selected without replacement from the entire training set. For each query molecule $m_q$, that is an element of some $\mathcal{M}_P$, a context molecule and its activity is randomly selected from all molecules with data against the same target, $(m_c, \pi_P(m_c))$,

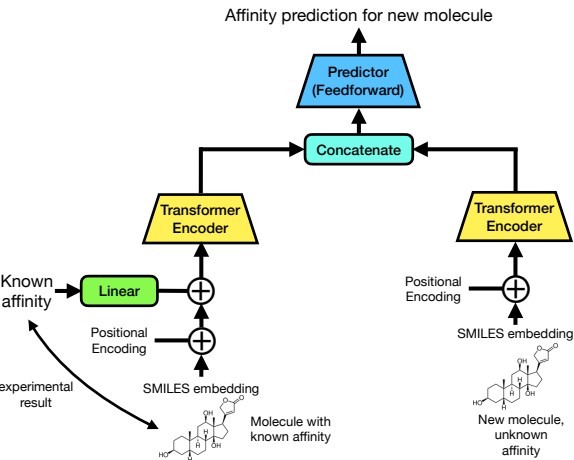

Figure 1: **Architecture overview.** The context encoder (left) receives the embedded SMILES tokens of the context molecule summed with a sinusoidal positional encoding and a linear projection of its known activity value. The query encoder (right), which has different weights, receives the embedded SMILES tokens of the query molecule summed with a positional encoding. A feedforward predictor network receives the concatenated outputs of each encoder and produces a final scalar activity prediction of the query molecule.

where $m_c \in \mathcal{M}_P$. When selecting the context compound, an additional constraint is placed on $\pi_P(m_c)$ such that $a \le \pi_P(m_c) < b$, where $a$ and $b$ are the lower and upper bound, respectively, of the manually set context activity range. By constraining $\pi_P(m_c)$ but not $\pi_P(m_q)$, OSLS can learn to score compounds with a wide range of activities using only weakly active compounds as context.

## 4 EXPERIMENTS

We test OSLS and baseline techniques on four applicable drug discovery tasks: (1) ranking and classification of compounds from BindingDB (Liu et al., 2007) with varying context compound affinities that are typical of screening results. (2) predicting the activities of anti-cancer drugs against a wide array of tumor cell lines (Barretina et al., 2012) using the activity of a single drug against each cell line as context for activity prediction of other drugs. (3) simulating a drug repositioning task using the natural ligand estradiol and a set of FDA-approved drugs (Selleck), and testing for the retrieval of diverse compound scaffolds that are already known to bind the target of estradiol. (4) use of OSLS as an objective function to guide molecular generative modeling using a single known ligand. Finally, we measure the impact of various ablations on the OSLS architecture.

Additional results from the adaptation of OSLS to the few-shot setting, following the experimental setup of Lee et al. (2022), are reported in Appendix A. Dataset and preprocessing details are reported in Appendix B, and details on the implementation of OSLS are reported in Appendix C.

### 4.1 BASELINES

We test OSLS against the following baselines, whenever applicable to a given task. Details on the training and implementation of baselines are given in Appendix D.

- **OSLS-MLP.** Similar architecture to OSLS except using Morgan fingerprints to represent molecules connected to a multilayer perceptron (MLP) instead of a Transformer. Loosely related to Koch et al. (2015), except we untie the two encoders and predict a continuous target. The fingerprint of the context compound is multiplied with its known activity value to encode context, a process inspired by POT-DMC (Godden et al., 2004).

- **Tanimoto similarity.** Molecular structure-based similarity measure based on vector similarity between two binary fingerprints. In this paper, we use Morgan fingerprints (Rogers & Hahn, 2010) with a radius of 2. This measure of molecular similarity is commonly used in practice.

- **MaLSTM.** Siamese neural network to embed molecular SMILES sequences using an LSTM network followed by a vector similarity measure (Mueller & Thyagarajan, 2016).

- **Graph Matching Network.** Classifies if two graphs are similar by jointly reasoning using a cross-graph attention mechanism (Li et al., 2019).

- **MetaDTA.** Applies attentive neural processes to the few-shot regression of continuous binding affinity values (Lee et al., 2022).

- **OpenEye Rapid Overlay of Chemical Structures (ROCS).** Computes overlap between 3D molecular representations, using both pure 3D shape ("ShapeTanimoto") and 3D shape combined with charge information ("TanimotoCombo") (Kearns & Pande, 2016).

### 4.2 COMPOUND RANKING AND CLASSIFICATION

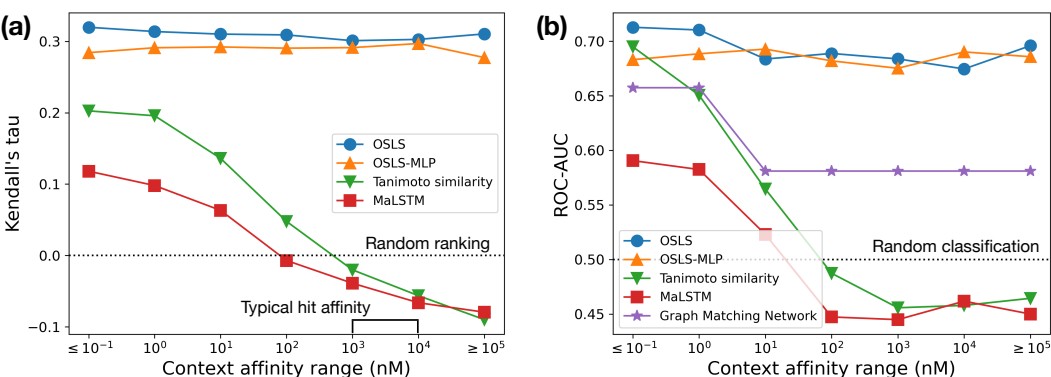

Figure 2: **Comparison of compound ranking and classification techniques at varying context compound affinities.** (a) mean Kendall's tau correlation across 40 test targets between real and predicted affinity among all compounds with data available against each target. Context compounds are taken at random from the set of compounds with activity data against each target with an affinity in the range specified on the x-axis. Marked is the most common range of hit affinities following a HTS campaign (Freire, 2015; Hughes et al., 2011). (b) mean ROC-AUC of classification ability across 40 test targets with the context compound affinity range on the x-axis, with true positives being defined as compounds with $< 10$ nM affinity against their target, and true negatives otherwise.

We compare `OSLS` with baselines on scoring compounds from a library, while varying the affinity of a context compound that is analogous to a hit from a high-throughput screen. We use a set of 40 unseen test targets from BindingDB, and aim to score all compounds that have experimental activity data against each target. For each query compound, a context compound is taken at random from all compounds against the same target within the defined affinity range (specified on the x-axis). For all methods, we train a separate model for each context affinity range. After scoring, we measure performance in ranking and classification, both common tasks in drug discovery.

Figure 2 left reports the mean rank correlation across all targets between true compound binding affinity and the score produced by each method. For the two similarity-based methods, Tanimoto similarity and MaLSTM, scoring was done to the context compound, i.e. the highest scored molecules are those most similar to the context compound. As expected, the ability of these techniques to rank compounds by their true affinity value declines rapidly as the context compound becomes less potent, reaching random ranking around the micromolar range ($10^3$-$10^4$ nM), which is the typical top affinity obtained from a high-throughput screening campaign (Freire, 2015; Hughes et al., 2011). In contrast, the performance of `OSLS` and **OSLS-MLP** is relatively strong, and strongest for `OSLS`, throughout all context compound affinity ranges.

We also use the scores produced by each method in a classification task, where query molecules are considered positives if the affinity to their respective target is $< 10$ nM, and negatives otherwise, to

reflect the typical desired affinity of lead optimization projects (Hughes et al., 2011). Figure 2 right shows the mean binary classification ability for each method across all test targets, as measured by ROC-AUC. In addition to the previous baselines, we also include Graph Matching Network (GMN), which is only applicable to the binary classification task. We trained and tested two different GMN models, one with a positive and the other a negative context compound, so the reported performance is identical across all affinity ranges that were included in each model's training set, as we used the same range for testing ($< 10$ and $\geq 10$ nM for the positive and negative context model, respectively). As shown, the performance of most methods drop as the context compound becomes less potent, although `OSLS` mostly retains performance even as the context compound is no longer drawn from the active range. Additionally, `OSLS` shows the highest ROC-AUC compared to baselines across most context affinity ranges, especially the low nM ranges.

Besides `OSLS` and OSLS-MLP, no other baseline can take advantage of continuous activity values, seemingly limiting their performance. Specifically, `OSLS` outperforms all baseline methods across most context affinity ranges, especially at the most common and applicable low micromolar range.

### 4.3 SCORING ANTI-CANCER COMPOUNDS

Chemotherapy involves the targeting of specific tumors with specialized drugs. Tumors are often profiled by genetic sequencing and the subsequent identification of cancer-causing mutations (Malone et al., 2020), but it is often not clear which drug will be active against a given mutation (Kalamara et al., 2018; Tsimberidou, 2015). One alternative approach to tumor profiling, that is already undertaken clinically, involves screening a small set of drugs against a patient-derived tumor culture, yielding continuous activity values for each drug (Letai, 2017; Wong et al., 2021).

Most cancers involve very specific mutations, but high-throughput screening (HTS) against any one mutation is often not economically or biologically feasible (Letai, 2017). One-shot learning techniques, however, can potentially extrapolate small sets of clinically obtained drug data to much larger HTS-scale libraries, expanding both clinical therapeutic options and making viable the evaluation of many preclinical drug candidates against highly specific tumors.

Towards this end, we evaluated the performance of `OSLS` and OSLS-MLP in scoring

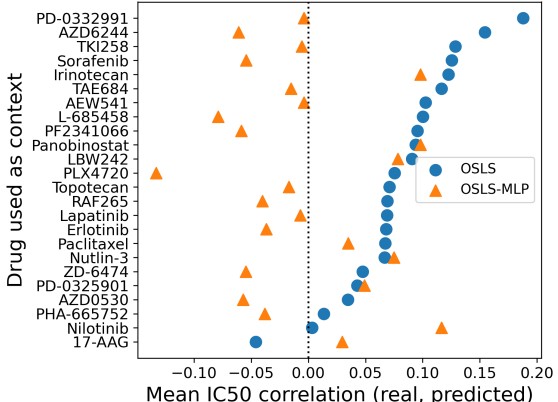

Figure 3: **Scoring drugs from the Cancer Cell Line Encyclopedia.** The x-axis shows the mean Pearson's $r$ correlation coefficient between real and predicted drug activity, as measured by normalized IC50 values, across a wide array of cancer cell lines. The y-axis shows the single context compound used for prediction, where its activity against each cell line informs the scoring of all other compounds against that line. Compounds are ordered by the correlation coefficient from `OSLS`, to show which compounds are most informative when supplied as context.

anti-tumor drugs against tumor cell lines from the Cancer Cell Line Encyclopedia (CCLE, Barretina et al. (2012)). Figure 3 reports the mean correlation across all cell lines (x-axis) between the real and predicted IC50 values of drugs with activity data against each line. We repeated the scoring using different context drugs (y-axis), which were supplied to both methods along with their respective activity values against each cell line. For each context drug, the reported correlation coefficient is computed using activity values normalized to have a mean of 0 (for both real and predicted values) to account for baseline differences in drug activity and capture only tumor-specific activity differences.

As shown, `OSLS` displays some level of predictive power across all but one context drug. OSLS-MLP, however, has less or no predictive power for most context drugs. Models were trained on BindingDB data only, and not the CCLE, suggesting that the observed predictive power of `OSLS` would transfer to other compounds beyond those tested here, including other clinical drug options or even preclinical candidates. Additionally, models were trained on binding affinity values, which,

while presumably related to the present IC50 values, indicates the potential for transfer learning across different activity types mediated by an underlying binding mechanism.

We only compared with one baseline on this task as many of our previously used baselines are not applicable. For instance, similarity-based methods will yield identical similarity values between the tested drugs regardless of the experimentally determined activity value, meaning they will have no predictive power. Binary classification methods are also not applicable, because defining any binary cutoff among the relatively smoothly distributed activity values is not necessarily meaningful, and binary outputs would also not produce meaningful correlations to the continuous ground truth. Therefore, only one-shot activity prediction methods were applicable to this task, and OSLS shows far superior performance to OSLS-MLP, which makes close to random predictions. Our results suggest that OSLS may be able to complement or replace existing tumor profiling techniques, and significantly reduce the costs associated with scoring a large compound library against specific tumor profiles, offering both clinical and preclinical utility.

## 4.4 SCORING FOR DRUG REPOSITIONING

Drug repositioning is the application of existing drugs to new indications, promising to avoid expensive drug development and approval while still offering needed new treatments (Pushpakom et al., 2019). Many computational techniques have been proposed for predicting the activity of existing drugs against novel targets, including chemical similarity-based techniques (Jarada et al., 2020). We simulated a drug repositioning task to the estrogen receptor (using models trained on BindingDB, where the estrogen receptor was excluded from the training set) by scoring all FDA-approved drugs by predicted activity using estradiol as the context compound, a natural ligand to the estrogen receptor. We chose the estrogen receptor, which has many existing drugs, to compare the ability of each technique to rank these drugs, which we knew beforehand, highly in comparison to all others.

Table 1: **Drug repositioning to the estrogen receptor.** Each technique was used to rank all 2,617 FDA-approved drugs using estradiol (a natural ligand to the estrogen receptor) as the context compound. The corresponding ranking (out of 2,617) of each drug which is known to bind the estrogen receptor (taken from Wishart et al. (2006)) is shown (lower is better).

| Drug | OSLS | Tanimoto similarity | ROCS (TanimotoCombo) | ROCS (ShapeCombo) | MaLSTM | OSLS-MLP |
|------|------|---------------------|----------------------|-------------------|--------|----------|
| Estrone | 113 | 6 | 4 | **3** | 5 | 223 |
| Raloxifene | **10** | 531 | 519 | 1281 | 1898 | 35 |
| Fulvestrant | 25 | **12** | 18 | 1191 | 95 | 54 |
| Tamoxifen | 214 | 1434 | 1527 | 1562 | 2493 | **198** |
| Bazedoxifene | 1023 | 353 | 684 | 1380 | 1162 | **239** |
| Toremifene | 887 | 1562 | 1527 | 1568 | 2576 | **562** |
| Ospemifene | 1077 | 2283 | **941** | 1469 | 2277 | 1498 |

Table 1 reports the rankings produced by each technique of all estrogen-binding drugs (taken from Wishart et al. (2006)) among 2,617 total FDA-approved drugs (obtained from Selleck). The steroidal drugs estrone and fulvestrant are ranked highly by all methods, as they are structurally similar to estradiol, which is also a steroid. For the other non-steroidal drugs, which are structurally very different from estradiol, most similarity-based methods fail, while the novel one-shot scoring techniques (OSLS and OSLS-MLP) rank these drugs relatively highly. These results suggest that one-shot learning methods are more capable of "scaffold hopping," which is the ability to identify compounds that are structurally distinct from a known compound but have similar activity.

## 4.5 GUIDING MOLECULAR GENERATION

Many generative models have been proposed to design novel compounds against a given target (e.g. Jin et al. (2018); Luo et al. (2021); Jin et al. (2020); Xie et al. (2021); Zhou et al. (2019); Zang & Wang (2020); Eckmann et al. (2022)). All generative models employ an objective function to score suggested ligands, which, when attempting to generate binders against a target, is typically docking software (Spiegel & Durrant, 2020; Eckmann et al., 2022) that requires target information. When the target is unknown, in the case of phenotypic or ligand-based drug discovery, some or all of these existing methods are not applicable. To the best of our knowledge, in the common case of

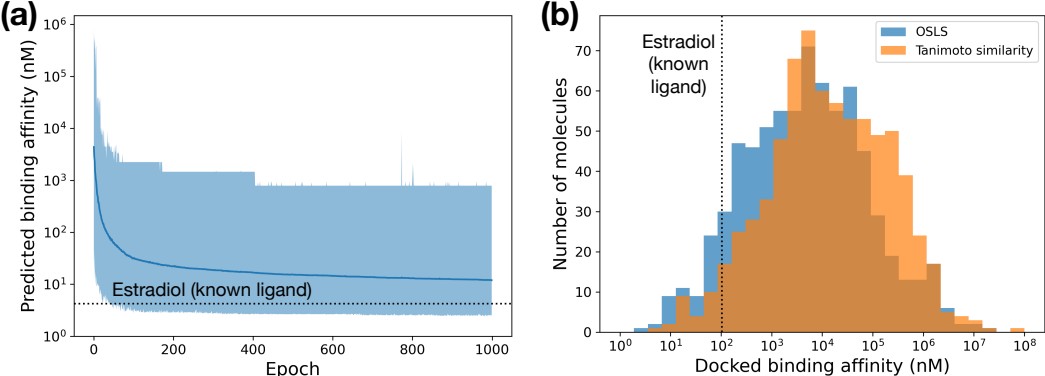

Figure 4: **Molecular generation using `OSLS` as an objective function.** (a) mean (solid line), minimum, and maximum predicted binding affinity of 256 variational autoencoder-generated molecules optimized with `OSLS` as an objective function (lower is better). The known affinity of estradiol, the natural ligand being used as a context compound, is also shown. The x-axis shows the number of gradient descent steps in the optimization process. (b) distribution of AutoDock-GPU scores among all valid generated compounds (lower is better), with the docked score of estradiol also shown.

*de novo* drug design based on one or a small number of hits from phenotypic screening (Swinney & Anthony, 2011), existing generative models cannot be applied, and is what we seek to address.

After training on BindingDB, we applied `OSLS` as an objective function in a variational autoencoder (VAE, Kingma & Welling (2013))-based generative model, using a single known compound (estradiol, a natural ligand) and its measured affinity to the estrogen receptor (a well-studied and disease-relevant target, which was excluded from the training set) as context, and seek to generate new molecules with higher affinity to the target than estradiol. We slightly adapt `OSLS` to allow end-to-end differentiation and combine it with a trained VAE, similarly to previously proposed generative models (Luo et al., 2021; Jin et al., 2018; Zang & Wang, 2020; Eckmann et al., 2022), and then backpropagate the output affinity prediction to the latent space to generate new molecules with higher predicted affinity (for more details about the training and method, see Appendix E).

Figure 4 left shows the mean, minimum, and maximum affinity prediction from `OSLS` among 256 molecules over 1,000 epochs of backpropagation. As shown, `OSLS` successfully guides generation towards compounds with higher predicted affinity, with some even surpassing that of estradiol. To further validate the generated compounds, Figure 4 right shows the distribution of generated compound affinities predicted by AutoDock-GPU to the estrogen receptor (PDB ID 1ERR, details in Appendix E) (Santos-Martins et al., 2021), a docking program that uses full target information. Encouragingly, many generated compounds surpass the docking-predicted affinity of estradiol. Note that docking, including AutoDock-GPU, is generally a poor predictor of actual affinity (Pantsar & Poso, 2018), so is not necessarily more accurate than the predictions of `OSLS` itself. We included the Tanimoto similarity baseline, which consists of a simple differentiable surrogate model of Tanimoto similarity to estradiol, to approximate the typical similarity search-guided molecular generation of medicinal chemists (see Appendix E for further details). As no other baseline was readily differentiable, we did not include any beyond Tanimoto similarity.

Table 2 reports further metrics of generated molecules, and compares with other generative models that employ target information (Xie et al., 2021; Olivecrona et al., 2017; Eckmann et al., 2022). Out of 256 molecules generated by each method, only molecules with a docked affinity better than that of

Table 2: **Comparison of generated molecules**. Quality metrics (see Appendix E) of generated compounds with higher docked affinity than estradiol. We report the best docked affinity as $K_d$ in nanomoles/liter, the number of Butina clusters, and the diversity and novelty scores as described in Jin et al. (2020).

| Method | Best (nM) (↓) | # clusters (↑) | Diversity (↑) | Novelty (↑) |
|---|---|---|---|---|
| REINVENT | 0.98 | 57 | 0.866 | 0.98 |
| MARS | 17.3 | 4 | 0.893 | 1.00 |
| LIMO | 0.31 | 57 | 0.874 | 1.00 |
| Tanimoto | 4.5 | 23 | 0.862 | 0.81 |
| OSLS | 2.4 | 42 | 0.862 | 1.00 |

estradiol were included in metric calculations (full details on the metrics used are available in Appendix E). While molecules generated by `OSLS` are surpassed in most metrics by generative models that harness target information, they are still of relatively high docked affinity. It is encouraging that such compounds were generated using only a single context compound, suggesting that `OSLS` may be used successfully for lead optimization in phenotypic drug discovery. In contrast to the Tanimoto similarity baseline, `OSLS` generates a high structural diversity of molecules with desirable affinity, which is of high utility for drug discovery.

## 4.6 Ablation Study

We report performance metrics of model variations in Table 3. We measure the average Kendall's tau and ROC-AUC (for the binary identification of $< 10$ nM compounds) across 40 unseen targets in BindingDB, using $< 1$ nM context compounds. We tested SMILES, SELFIES (Krenn et al., 2020b), and SMILES Pair Encoding (SPE; Li & Fourches (2021)), which are string-based molecular representations. "Continuous" means that a continuous activity value was provided as context, "Binary" that the activity was thresholded at 10nM to either 1 or 0, and "No context" that no context was provided at all. "Linear" and "Non-linear" (cubic) projections were also tested to encode scalar activity values.

Table 3: **Ablations**. Blank entries represent no variation from the base model (top).

| Molecule representation | Context provided | Projection type | Kendall's tau | ROC-AUC |
|---|---|---|---|---|
| **SMILES** | **Continuous** | **Linear** | **0.3198** | **0.7128** |
| SELFIES | | | 0.2901 | 0.6820 |
| SPE | | | 0.2255 | 0.6524 |
| | Binary | | 0.2107 | 0.6881 |
| | No context | | 0.3024 | 0.6842 |
| | | Non-linear | 0.3177 | 0.7087 |

## 5 Discussion and Conclusions

We present `OSLS`, a one-shot learning model for ligand scoring utilizing a Siamese-inspired neural architecture with two Transformer encoders without tied weights, a novel positional encoding-like mechanism, and a final prediction head. After training on BindingDB, a large dataset of drug-target affinity data, we show that `OSLS` surpasses both traditional and recent baselines on a variety of relevant tasks involving one-shot ligand scoring on unseen targets. Promisingly, `OSLS` is able to correctly score compounds that are of higher activity that the provided context compound, which is shown to be valuable in compound retrieval, drug repositioning, and generative tasks.

In contrast to previous work, we utilize continuous, rather than binary, activity values, which seems to be beneficial even for binary classification tasks (Figure 2, Table 3). Continuous values are also useful for ligand scoring following screening, where hits are active but of mild potency, in which case binary methods cannot distinguish between mildly and highly active compounds. Finally, continuous values are highly applicable to some clinical and preclinical challenges, such as those in cancer therapy, where *how* active a compound is can be highly relevant (Section 4.3).

Our work also differs from previous work in that we focus on one-shot, rather than few-shot, prediction. The number of hit compounds obtained from a high-throughput screen (HTS) increases with time and resources (Attene-Ramos et al., 2014), so being able to terminate screening after only one identified hit (which would be used as the context compound for `OSLS`) would greatly reduce costs. HTS is also usually followed by lower-throughput confirmatory assays, which in practice can result in as a few as one active compound with a measured affinity (Zhu et al., 2013), meaning our approach is of much broader applicability.

`OSLS` may also be applied to phenotypic drug discovery, where the target is unknown and hits are determined by observing a cellular response. Even if there are multiple hits resulting from such a screen, many will not mediate a response through the desired cellular mechanism of action, and so scoring additional compounds based on these hits is not useful (Vincent et al., 2020). As typically only one or a very small number of hits induce a response through the desired mechanism of action, scoring of further active compounds using only a single hit (presumably, the hit with the desired mechanism of action) gives maximum flexibility for and applicability to phenotypic drug discovery.

## 6 REPRODUCIBILITY STATEMENT

Anonymized source code for the base `OSLS` model is provided in the supplementary materials, as well as helper functions to load pretrained models and perform inference using raw SMILES strings. Scripts are also provided to preprocess data from BindingDB and prepare it for training. Descriptions of preprocessing steps undertaken for each dataset are provided in Appendix B. Implementation details for `OSLS` are provided in Appendix C, and descriptions of how each baseline was implemented are provided in Appendix D. Finally, implementation and baseline details for the generative modeling experiments are provided in Appendix E.

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

Table 4: **Comparison of few-shot learning methods for continuous binding affinity prediction.** Results for all methods besides `OSLS` taken from Lee et al. (2022). While code was not made available, we attempted to as closely match the reported experimental setting as possible. $r_m^2$ is reported for both a context set size of 10 and 20 ligands, and is defined as $r^2 \cdot (1 - \sqrt{r^2 - r_0^2})$, where $r^2$ and $r_0^2$ are Pearson's correlation coefficient with and without intercept, respectively (Lee et al., 2022).

| Method | $r_m^2$ (10 ligands) | $r_m^2$ (20 ligands) |
|---|---|---|
| DeepDTA | 0.299 | 0.360 |
| MolTrans | 0.363 | 0.410 |
| MetaDTA(I) | 0.468 | 0.383 |
| MetaDTA(T) | 0.374 | 0.433 |
| OSLS | **0.523** | **0.543** |

## A  FEW-SHOT LEARNING RESULTS

Although we focus on one-shot learning, few-shot learning is a related problem that is important in the literature and finds applications in some forms of drug discovery. Few-shot learning involves predicting the binding affinities of query ligands to a target not seen in training, using multiple known support ligands and their affinities to the same target. We seek to reproduce the few-shot binding affinity regression task previously described in the literature as closely as possible, using data from a test split of BindingDB targets with more 20 compounds each, a support set randomly drawn over the entire binding affinity range for a given target, and aim to predict the binding affinity of randomly chosen query ligands from the same target (Lee et al., 2022).

To make `OSLS` compatible with few-shot learning, we encode each context compound and its affinity using the context encoder, and then apply an additional "context attention" encoder over all encoded context vectors that we average to produce a final context vector. This "context attention" module is a Transformer encoder with 8 attention heads, 4 layers, and a final linear layer that takes each context vector as a single element of the input sequence. We do not apply positional encoding, as the ordering of each ligand in the sequence is not relevant. As shown in Table 4, `OSLS` produces binding affinity predictions that are significantly more correlated to the true affinity than baseline methods, for both a context set of size 10 and 20.

## B  DATASET DETAILS

For training, we use BindingDB (Liu et al., 2007), a dataset of 2.6 million drug-target binding affinities, for its high data quality and broad coverage across many protein targets. We exclude very small or very large molecules, defined as less than 10 atoms or more than 70. We record affinities in nanomolar units from the $K_d$ column if available, and if not, $K_i$. If neither value is available, we discard the molecule. When affinity is expressed as an upper or lower bound, we take the bound itself as the known affinity. Then, we exclude all protein targets, and the compounds therein, if there are less than 10 compounds with data against that target or if there are no compounds in any one of the binding affinity ranges shown in Figure 2. Finally, we perform a target split, where we set aside 40 targets for testing only, and train on the remaining targets, with target being defined by protein sequence. We transform all binding affinity values using the base-10 logarithm, as binding affinity often spans several orders of magnitude. All experimental settings involve compounds only in the set-aside testing set of targets. While the train/test split is otherwise random, we make sure to include the estrogen receptor in the test split, as Section 4.4 and 4.5 use the estrogen receptor in the contained experiments.

The models trained on BindingDB were also tested on a number of different datasets. The Cancer Cell Line Encyclopedia (Barretina et al., 2012) consists of interaction data of 24 drugs against a wide array of 479 patient-derived cancer cell lines. For the results reported in Section 4.3, we use the dataset reported in Table S11 of Barretina et al. (2012), and extract IC50 measurements for each drug measured against each cell line. We exclude compounds with less than 10 or more than 70 atoms, and cell lines with less than 2 drugs with measured activity. For the results reported in

Section 4.4, we use the Selleck dataset of FDA-approved drugs (Selleck). For drugs that are reported with a salt compound, we remove the salt from the SMILES string. We also exclude molecules based on the same size constraints as above.

## C  IMPLEMENTATION DETAILS

**Hyperparameter tuning.**  We tuned hyperpameters of `OSLS` once for the task described in Section 4.2 using a $< 1$ nM context compound, and used the same hyperparameters for all other context affinity ranges and subsequent tasks. We measured model performance by Kendall's tau after $2^{15}$ training steps. The following hyperparameters were entered into a grid search, and the observed best value for each hyperparameter is bolded: `learning rate={1e-4, 5e-4, 1e-5, `**`5e-5`**`, 1e-6 5e-6}, batch size={512, `**`1024`**`, 2048}, d_model={256, `**`512`**`}, n_heads={8, 12, `**`16`**`}, n_layers={`**`4`**`, 6}`

**Training.**  We used an Adam optimizer with a base learning rate of $10^{-5}$ and 128 steps for learning rate warmup, and then cosine annealed the learning rate to 0 over $2^{15}$ steps. We applied dropout ($p = 0.1$) over the scalar affinity projection, the token embeddings for both encoders, and at each layer in the predictor network. All models were trained on a server using a single NVIDIA GTX 3080 GPU.

## D  BASELINE DETAILS

**OSLS-MLP.**  We implemented and trained a simpler but `OSLS`-like architecture to show the added benefit of the more complex, Transformer-based `OSLS` model. This simpler model is loosely similar to Koch et al. (2015), except we untie the two encoders so that the context and query molecule may be encoded differently (which, like `OSLS`, we call the context and query encoders), and perform regression instead of classification. In particular, the context must include a molecule representation as well as activity information, but not the query, so using the same network to encode both would be non-optimal. To represent molecules, we use Morgan fingerprints with 1024 bits and a radius of 2, and both encoders as well as the predictor (the so-called "distance layer" in Koch et al. (2015)) are feed-forward ReLU-activated networks with linear output. The distance layer consists of only a single linear layer, similarly to the previously proposed Siamese network by Koch et al. (2015). For the context molecule, we multiply the binary fingerprint vector of the compound with its activity value (the base-10 log of $K_d$ or $K_i$), so that each non-zero element of the vector takes on the activity value. This is inspired by POT-DMC (Godden et al., 2004), although we do not perform an additional element-wise normalization step. The network is trained in a similar way to `OSLS`, with mean squared error loss between the real and predicted binding affinity of the query molecule at the output of the distance layer. We trained the model until convergence using the Adam optimizer. For hyperparameter optimization, we selected the highest Kendall's tau on the previously described ranking task with a context molecule of $< 1$ nM affinity over a grid search of the following hyperparameters (the best hyperparameters are bolded): `encoder layers: {2, `**`3`**`, 4}, encoder layer width: {`**`1024`**`}, batch size: {64, `**`128`**`, 256}, learning rate: {1e-4, `**`1e-5`**`, 1e-6}`

**MaLSTM.**  We implemented and trained the MaLSTM model (Mueller & Thyagarajan, 2016) to predict binding similarity using the SMILES tokens of the context and the query molecules. As the context and query encoders must share weights, we did not incorporate binding affinity information for the context molecule, and we trained the model only to predict the similarity in binding affinity between two molecules. To do this, we used single-layered LSTM blocks, as reported in the original paper, that received learned embedded SMILES tokens as input. We computed similarity using the last hidden state of both the query and context encoders ($a$ and $b$, respectively), terminating at the end of each SMILES sequence, as defined by the exp of the L1 norm of the difference between $a$ and $b$: $\exp\left(\|a - b\|_1\right)$. We used the mean squared error between the predicted similarity and the true similarity, which we define as $\exp\left(-\mathrm{abs}(\pi_P(m_c) - \pi_P(m_q))\right)$, where $\pi_P(m_c)$ and $\pi_P(m_q)$ are the true activity values of the context and query molecules, respectively, to the target $P$. The selection of query and context molecules during training is identical to the process described in Section 4.2. We trained the model until convergence using the Adam optimizer. Simi-

larly to hyperparameter optimization for our model, we selected optimal hyperparameters based on the highest Kendall's tau on the previously described ranking task using a context ligand of $< 1$ nM, and then used the same hyperparameters for all subsequent tasks. For hyperparameter optimization, we performed a grid search over the following set of hyperparameters (the best hyperparameters are bolded): `batch size={256, 512, `**`1024`**`}`, `embedding dim={64, `**`256`**`, 512}`, `hidden dim={`**`50`**`, 100, 200, 500}`, `lr={1e-3, `**`1e-4`**`, 1e-5}`

**Graph Matching Networks.** We trained Graph Matching Networks for binary prediction of molecular similarity using the implementation provided at `https://github.com/Lin-Yijie/Graph-Matching-Networks`. For the binary classification task described in Section 4.2, we trained and tested two separate models, one using a positive context molecule and the other a negative. Specifically, while we used a similar context and query sampling procedure as described in Section 4.2, we sampled the entire "active" range ($< 10$ nM) to generate context molecules for training the first model, and the entire "negative" range ($\geq 10$ nM) for training the second model. We did not provide the model with any continuous affinity data, as the model is designed to operate only on two graphs with no additional information. As a training signal for the first model, we assigned a "similar" label if the query molecule was active (as the query and context molecules would both be active in that case), and otherwise "dissimilar". For the second model, we assigned a "similar" label if the query molecule was inactive, and "dissimilar" otherwise. Molecules were represented as NetworkX graphs, with bond type represented as an integer as an edge feature, and atom type as a one-hot encoded node feature. For the first model, we reported ROC-AUC in a similar way as other baselines, but for the second model we reported $1 - AUC$, as a dissimilar prediction to a non-active context molecule would actually indicate an active query. We followed a similar hyperparameter optimization scheme as described previously, except we used the highest ROC-AUC score instead of Kendall's tau on the test set to evaluate hyperparameter quality, as Graph Matching Networks are not applicable to ranking. We performed a grid search over the following set of hyperparameters (the best hyperparameters are bolded): `graph_representation_dim={`**`128`**`, 256}`, `prop_layers={`**`5`**`, 10}`, `lr={1e-3, `**`1e-4`**`, 1e-5}`, `loss={`**`margin`**`, hamming}`, `batch_size={10, `**`20`**`, 40}`

**OpenEye Rapid Overlay of Chemical Structures (ROCS)** OpenEye Scientific Software offers Rapid Overlay of Chemical Structures (ROCS; Hawkins et al. (2006)), a program used here to determine shape and electronic similarities between estradiol and drugs in the Selleck FDA-approved drug database (Selleck). The similarity metrics used are *ShapeCombo*, which scores compounds against estradiol with Gaussian spatial similarity estimations and *TanimotoCombo*, which combines equal contributions of Gaussian spatial similarity estimations and the *ImplicitMillsDean* force field's (Mills & Dean, 1996) high-level chemical similarity estimations. To perform the ROCS analyses, 3D conformer generation of compounds in the Selleck database was required. The conformer generation was achieved using OpenEye's OMEGA software (Hawkins et al., 2010), providing the lowest-energy conformations which are then analyzed by ROCS. The result of ROCS analysis is a scoring and ranking of Selleck database molecules by spatial and electronic similarity to estradiol. We only used ROCS as a baseline in the drug repositioning task, as conformer generation took around a second per compound, which was too slow for the task in Section 4.2 that involved around 2 orders of magnitude more compounds.

# E    GENERATIVE MODELING

We selected a variational autoencoder (VAE, Kingma & Welling (2013)) as our generative model due to its relatively easy training and ability to take advantage of differentiable objective functions. We trained a simple feedforward VAE to reconstruct compounds from BindingDB (the same set as used in other tasks) using byte-pair encoded SMILES tokens as a molecular representation. After experimentation with other string-based molecular representations (raw SMILES tokens and SELF-IES; Krenn et al. (2020a)), byte-pair encoding of SMILES strings (Li & Fourches, 2021) generated the most reasonable molecular samples from sampling random locations in the latent space (i.e. the fewest number of very large rings and uncommon atom and bond configurations). The byte pair-encoded SMILES tokens for each molecule were embedded into 512 dimensions and then padded to the maximum length, which was then flattened such that the input and output of the VAE had dimension $max\_len \cdot embedding\_dim$. After the output of the VAE decoder, we unflattened the sequence

into a $max\_len \times embedding\_dim$ matrix and applied softmax over each position in the sequence, so that the summed probability of all possible tokens at each position in the sequence was 1. The encoder and decoder of the VAE were mirrored, both having 3 1024-dimensional hidden layers with ReLU activations and batch norm before the activation. We trained the VAE until convergence using the Adam optimizer with a learning rate of 1e-4. The loss function was negative log-likelihood loss between the true and reconstructed output plus a KL-divergence term on the latent space of 1024 dimensions. During training, we warmed up the weight of the KL-divergence loss starting from 0 to 1 at epoch 1,000 to avoid mode collapse.

Separately, we trained `OSLS` on BindingDB using the same hyperparameters as previously described except using a random context compound from the entire binding affinity range, and with byte-pair encoded SMILES instead of raw SMILES tokens so that the VAE and `OSLS` can be stacked, following Eckmann et al. (2022). While the final trained model had worse performance than the raw SMILES version (see Section 4.6), it was sufficient for this purpose. To make `OSLS` fully differentiable so that it can be used as an objective function for the VAE, we applied Gumbel softmax instead of regular softmax at the output of the VAE decoder, so that the output of the VAE was fully discretized, but still allowed for differentiation through the "straight-through" technique, where the output of Gumbel softmax is fully discretized in the forward pass but an approximate version is used for the backward pass (Jang et al., 2016). With a set of discretized tokens, we were then able to perform matrix multiplication on the one-hot encoded tokens (produced by Gumbel softmax) with the weights of the embedding layer, which produces the same embeddings as the typical indexing-based method, but in a differentiable manner. These embeddings were then fed into the encoders of `OSLS`, and then the prediction head, making for a fully differentiable forward pass from VAE latent space to output activity prediction. Similarly to Eckmann et al. (2022), we attached the input of the differentiable `OSLS` architecture to the output of the VAE decoder, and propagated the gradients of the `OSLS` output layer back to the latent space, altering the latent space in such a way as to generate compounds with higher predicted activities.

**Tanimoto similarity baseline.** We trained a convolutional neural network to approximate Tanimoto similarity between random compounds from the entire BindingDB dataset and estradiol, to simulate a similarity-based molecular generation process. This surrogate model consisted of 3 1-D convolutional layers, that take a sequence of 128-dim embedded SMILES byte-pair encoded tokens as input, with a kernel size of 3 and stride of 1, ReLU activations, and a final 1-D max pooling layer. The output of the convolutional layers was flattened, then fed to a 3-layer ReLU-activated feedforward network with batch normalization, which produced a final sigmoid-activated prediction of the Tanimoto similarity. The network was trained until convergence using the Adam optimizer with a learning rate of 1e-3 with mean squared error loss between real and predicted Tanimoto similarity to estradiol, and the embedding layers were made differentiable, and hence able to be attached to the VAE decoder, using the matrix multiplication trick described in the previous paragraph. There was a strong correlation between the real and network-predicted Tanimoto similarity on a held-out test set ($r^2 > 0.9$), meaning the network was an effective surrogate model for Tanimoto similarity. As the hyperparameters of `OSLS` were not optimized specifically for the generative task, and the trained model already had close to optimal performance, no hyperparameter optimization was performed for the Tanimoto surrogate model. Similarly to `OSLS`, we attached the Tanimoto surrogate model to the output of the VAE decoder and backpropagated its output back to the latent space of the VAE so that the generated molecules were of higher Tanimoto similarity to estradiol.

**Other baselines.** We compare the target-free generative models with REINVENT (Olive-crona et al. (2017); `https://github.com/MarcusOlivecrona/REINVENT`), MARS (Xie et al. (2021); `https://github.com/bytedance/markov-molecular-sampling`), and LIMO (Eckmann et al. (2022); `https://github.com/Rose-STL-Lab/LIMO`), all recently proposed generative models that can use full 3D target information. We ran each generative model until 256 valid molecules were generated, and then computed metrics on those molecules. We set the objective function for all models to be the docked energy to the estrogen receptor (as described in the "Docking details" paragraph). Default hyperparameters were used for all models, and pretrained models were used where applicable (the Prior of REINVENT, and the VAE of LIMO).

**Metrics.** We report the following metrics to evaluate the quality of generated molecules. For all methods, we only report these metrics on valid SMILES strings.

- **Best (nM)** the lowest generated docked affinity value in nanomoles/liter.
- **# clusters** the number of Butina clusters (Butina, 1999), where each cluster consists of molecules that have at least a 0.6 Tanimoto similarity to each other, computed using 1024-dimensional Morgan fingerprints with a radius of 2.
- **Diversity** one minus the average pairwise Tanimoto similarity between molecules. We use the same number of molecules across all methods to compute this metric as scores may be different across sets of molecules with different sizes even if the overall number of compound scaffolds, which are most relevant to drug discovery, are the same.
- **Novelty** the proportion of molecules that are less than 0.2 Tanimoto similarity to estradiol.

**Docking details.** We used AutoDock-GPU (Santos-Martins et al., 2021) to evaluate the molecules from generative modeling. A grid file of PDB ID 1ERR was produced by AutoGrid4, using the same binding site as estradiol. 10 random conformers of each ligand to score were then generated using obabel 3.1.0, docked against the grid file using the default AutoDock-GPU parameters, and then the best energy was taken from the 10 runs as the final docked energy.

