# OpenReview forum: "Target-Free Ligand Scoring via One-Shot Learning"
_ICLR.cc/2023/Conference — Submitted to ICLR 2023_

### Official Review · Reviewer_qHx9 · 2022-10-14

**Confidence:** 4
**Correctness:** 3
**Technical Novelty And Significance:** 2
**Empirical Novelty And Significance:** 2
**Recommendation:** 3

**Clarity, Quality, Novelty And Reproducibility:**

Quality:
The work surveys some existing methods for library screening well. However, it does not pay adequate attention to the existing few-shot and one-shot learning literature, in particular it does not fully and fairly explore the range of baselines in existence (although tested in other domains) for regression tasks in this setting, and rather focuses primarily on baselines where the performance is already expected to be poor.

Clarity:
Overall, the paper is clear and easy to read.

Originality:
The work makes a very minor architectural change to something that is very much like existing neural process architectures, although the reasons for the precise choices made are unclear and the results not compared against the alternatives, which makes it difficult to judge if this small change is making a significant difference to performance. The work is presented as "Siamese network inspired" which is not quite correct, and possibly leads to the work not being compared to the architectures to which it is actually more similar such as Protonets and neural processes.


**Strength And Weaknesses:**

Strengths:
* This particular configuration of modelling has not been seen before in the literature.
* The range of datasets and tasks explored is wide and a detailed analysis of the results is performed.
* The paper is in general clear and well-written.
* The model does appear to show promise for a range of tasks such as drug scoring, ranking, classification and as an objective in molecule generation.

Weaknesses:
* The authors refer to this as a Siamese-inspired or Siamese-like network. It is not, as there is no contrastive loss or comparative triplet loss or similar that forms a component of the model. This is somewhat misleading, as the only real similarity is the encoding of two molecules "simultaneously", which is similar to other methods such as neural processes, prototypical networks, etc.
* It is not truly clear why two separate transformer networks are required. While one must indeed process vectors where the scaling due to the scalar value of the context molecule has been provided, a justification, perhaps in the form of comparison to the case where the weights are tied, would be valuable.
* On a similar point, were the transformers weights tied, this model bears quite significant similarity to an adaptive neural process. The main difference is that the context molecule "label" is passed from the very start of the model, rather than merely being used to compute an MSE loss at the end. It would be helpful to discuss why this is seen as the correct approach. In addition, it would be possible to have a similar architecture where the scalar embedding is concatenated with or used to scale the post-transformer embeddings, prior to the final feedforward network. Did the authors try this? A discussion of this particular choice would be valuable.
* MetaDTA is presented as a baseline, but the results are not made available in the main body of the paper and the implementation used for results presented in the appendix is not described and is unclear. Given that it is listed as a baseline, it should be presented in the body of the paper alongside the other baselines. In addition, it is referred to as a few-shot only technique in the appendix, but there is no reason the context set cannot be reduced to the size of N = 1, in general, so it should be a suitable comparative baseline.
* All baselines are not presented together in Figures 4(a) and 4(b). It is possibly expected that in particular the Tanimoto and MaLSTM baselines might be weaker in Figure 4(a) in particular, and it would be useful to see results for the others mentioned.
* It is not clear why in the section on scoring anti-cancer compounds a neural process baseline such as MetaDTA is not used, as this is a few-shot/ one-shot regression task.
* The sentence (anti-cancer compounds paragraph 5) "similarity-based methods will yield identical similarity values between the tested drugs regardless of experimentally-determined activity value, meaning they will have no predictive power" is correct but confusing given the context. This is because many of the baselines in the list are presented as indeed being similarity measures but are being used to predict a unique score for each query molecule, in other parts of the paper (as a particular example, in Figure 2(a) MaLSTM and Tanimoto similarity are being used to predict affinity). Why is it not possible to predict a score in this new task with these baselines? If it is, they must be included in Figure 3. If not, then the paper needs to clarify this point in more detail, as currently there is not comparison in Figure 3 to any "external" method.
* Similarly, why is MetaDTA or another similar method excluded from Table 1. All baselines should be reported for fair comparison.
* In Figure 4 only Tanimoto similarity is given as a comparison objective in molecular generation. The claim is that no other model is readily differentiable, but it is not clear why this is the case, many of the baselines should be differentiable or have (as is claimed here for the tanimoto case) surrogates that are. In general this part of the paper is not clear, and does not seem well supported. The comparison are being made among other generative models rather than the objective function, which is the issue under investigation. The paper might be better to choose a single state of the art generative model and then investigate a broader range of objectives including the author's own. In addition, in Figure 4(a) there is no comparison even to the tanimoto baseline.
* The appendix lacks any details about MetaDTA, a description as thorough as those given for other models would be helpful.
* There exist a range of pretrained networks for molecular property prediction that could be incorporated in a neural process architecture to provide a robust baseline. Did the authors try anything like this?
* The work references related work in N-shot learning but does not try the range of methods referenced in the low shot setting, where they may be equally applicable.
* The work also suggests that the N-shot methods can only be used in binary classification settings, which is not correct. A trivial change to these architectures on a final layer and choice of a different loss create a regression model. It would be helpful to consider this section of the literature and make reference to it.



**Summary Of The Paper:**

This paper addresses the problem ligand scoring and identification of leads by screening libraries, tackling the specific problem of requiring an actual activity value or other numeric score (rather than classification) in the context of having only one labelled molecule. It approaches this problem by using an architecture that consumes a single context molecule and its label alongside a query molecule with transformer-based networks to learn the necessary functions.

**Summary Of The Review:**

The paper presents small architectural changes as compared to existing few-shot methods such as neural processes, but compares primarily against baselines not used in the broader one-shot and few-shot literature but rather similarity-based baselines used in only drug-discovery.  The proposed architecture is a small adjustment on some of the ideas from the one-shot and few-shot community and is not strongly justified nor are the slightly different alternatives explored. The experimental analysis does not fully compare against all baselines that the authors raise initially (although this list is not a good survey of the state of the art in the one-shot community). In particular it does not fully compare against those which are more complex than fingerprint similarity methods, and does not yet fully justify why that is the case.

---

### Official Review · Reviewer_SvkE · 2022-10-23

**Confidence:** 4
**Correctness:** 2
**Technical Novelty And Significance:** 2
**Empirical Novelty And Significance:** 2
**Recommendation:** 3

**Clarity, Quality, Novelty And Reproducibility:**

**Clarity**: generally clear, although some details of the experiments were a bit hard to guess

**Novelty/Originality**: the method is essentially a metric-learning method which doesn't seem that different from most other works in metric learning. I would say the novelty is incremental.

**Quality**: this work addresses a problem setting which I think is unrealistic, and justifies the method empirically using metrics which I was critical of. Therefore I do not think the quality of this work is particularly high.

**Reproducibility**: code is included, so I think reproducibility is good.

**Strength And Weaknesses:**

**Strengths**

- Writing is generally clear
- Given the problem setting, the method is reasonable (a neural-network based similarity)

**Weaknesses**

- Problem setting is unrealistic: I do not think predicting compound activity given just one active compound is a problem that has ever or will ever occur in drug discovery. As the authors mention in their introduction, initial hit compounds come from large parallel screening of compounds, which not only may produce multiple hits, but will also produce many non-hit compounds. By framing the problem as one-shot learning, the authors implicitly suggest discarding this information, which does not make sense to me and is not well-justified. I think this also makes their baseline methods unnecessarily weak: for example, Tanimoto similarity would not just be computed based on a single reference molecule, it would be computed based on all measured molecules (e.g. the inactive molecules too). This would be used to create a more complex ranking.
- Inappropriate metrics used: in Figure 2-3 the authors evaluated the methods based on the correlation between similarity and actual score across all molecules. This does not reflect how these methods will actually be used, which is just to find top compounds. Only the scores of the top few compounds are important; it does not matter if the correlation is low for very dissimilar molecules. In general I think the assumption of these retrieval-based methods is that the most similar compounds are likely to be active, _not_ that dissimilar compounds will be inactive. I think different metrics are required.
- Field of metric learning is not discussed in the related work. OSLS essentially seems like a metric learning method so I think these methods should be discussed.
- In the generative modelling baseline, I think that training a differentiable surrogate to Tanimoto similarity introduces an extra source of error which makes this baseline unnecessarily weak. I think the authors should find a different way to incorporate this information.

**Summary Of The Paper:**

The paper proposes one-shot ligand scoring (OSLS), a method to score potential drug candidates based on a single known active drug molecule. The method is essentially a Siamese neural network which takes in the active and query molecule to predict a score.

**Summary Of The Review:**

Despite being an interesting attempt at a relevant problem (drug-discovery), I think the authors target a very unrealistic variant of the problem, propose a method which seems like a fairly general metric learning method, and evaluate it in a way which I think is flawed. Together these things make me think that rejection is the appropriate choice.

---

### Official Review · Reviewer_ZuUo · 2022-10-23

**Confidence:** 5
**Correctness:** 3
**Technical Novelty And Significance:** 3
**Empirical Novelty And Significance:** 2
**Recommendation:** 3

**Clarity, Quality, Novelty And Reproducibility:**

The paper is overall well written and easy to read, with some exceptions. I am grateful that the authors provided the code, which helped to better understand the paper. That being said, the paper should be self-contained. There are bits missing that hinders the reproducibility of the paper.

The linear projection is not described anywhere in the paper. My understanding is that it produces a vector, where each element is the product of the scalar activity and a learned weight, and this vector is effectively broadcast across the input sequence dimension before adding it to the molecule embedding matrix. Is this correct? Either way, it should be explained more clearly.

The datasets should be better described, both in the paper and appendix. Perhaps a tabular summary would be helpful. BindingDB citation is outdated (Gilson, M. K. et al. 2016, BindingDB in 2015: A public database for medicinal chemistry, computational chemistry and systems pharmacology. Nucleic Acids). What version are authors using? When was the data obtained? How many drugs and targets are authors using?

The description of the training and testing sets in the paper is incomplete. How many drugs are used in training/testing? How many targets are used in training? I understand that authors are using one held out test set, but  are citing Feng et al, 2018, which uses a 5-fold cross validation.


**Strength And Weaknesses:**

The paper is well written and easy to read in most sections. The model is easy to comprehend, and the explanation is short and to the point. The results comparing to one shot approaches seem interesting and promising.

Scoring ligands based only on similarities of ligands, without information about the targets is an old approach. The use of information about targets has been shown to give better performance (Yamanishi et al. 2008, Prediction of drug–target interaction networks from the integration of chemical and genomic spaces). To support the utility and applicability of the proposed problem, authors could show a realistic scenario where there is no information about the protein targets. It would be interesting to see how OSLS compares to state of the art methods using target information. It would also help the paper to show any practical example of an application of ligand scoring where there is no information about the targets.

While the use of one shot approaches in ligand scoring can be theoretically justified, I am not convinced that there is any advantage of it over other n-shot approaches to the problem. My previous point also applies here, it would help the paper to show any practical example of an application where an n-shot approach is not applicable, or at least show any advantage of OSLS against n-shot methods. Table 4 in the appendix is a good start, but a more thorough analysis is needed.

The paper should be self-contained. More details in the next section.

In section 4.2, I would also like to see the AUPR, which is preferred in the literature for this classification task.

Section 4.4 is titled tries to show the use of OSLS in the drug-repositioning problem, but the task evaluated is drug target prediction (DTP). Drug repositioning is the prediction of drug disease associations. While DTP can be used in drug repositioning, the problems are not equivalent. If authors want to use OSLS in drug repositioning, they should change the experiment to show how OSLS can help to predict drug-disease associations. If the authors want to keep the experiment, they should show a systematic evaluation for DTP (for example “Wan, F. et al, NeoDTI: neural integration of neighbor information from a heterogeneous network for discovering new drug–target interactions. Bioinformatics 35, 104–111 (2019).”)


**Summary Of The Paper:**

The paper proposes One-Shot Ligand Scoring (OSLS) to predict activity of a compound to an unseen target based on a single context drug and its known activity to the given target, without any target features. They frame the ligand scoring as a one-shot problem, and argue for its utility over traditional similarity based scoring.

OSLS is a Siamese inspired neural architecture with a novel linear projection to model the activity of the context compound.
They show that OSLS outperforms both similarity based and deep-learning scoring methods in different settings and tasks related to drug development.


**Summary Of The Review:**

There are many missing details that hinder the paper's reproducibility, especially in the evaluation section.

Authors should try to justify the validity of their prediction setting more (for example, in which cases researchers have no information about protein targets) or show what advantages their approach has over other methods.

The evaluations proposed are not convincing.

---

### Official Review · Reviewer_3vLq · 2022-10-25

**Confidence:** 4
**Correctness:** 3
**Technical Novelty And Significance:** 3
**Empirical Novelty And Significance:** 3
**Recommendation:** 5

**Clarity, Quality, Novelty And Reproducibility:**

The introduction of the proposed model is clear.
The novelty is limited, as the transformer-based methods are common in this field.
Some experiment results do not support its effectiveness for molecule generation.

**Details Of Ethics Concerns:**

No ethics concerns.

**Strength And Weaknesses:**

Strengths:
   The authors explain the problem in detail and conduct various experiments, giving the reader the opportunity to look at the problem from a new perspective.

Weaknesses:

1) According to Figure 2 and Table 1, model with transformer architecture (which has much many parameters) does not outperform significantly that with MLP architecture (which has much less parameters). Is transformer architecture necessary?

2) According to Table 2, the proposed model did not show better performance than two baseline models. Maybe it is better to focus on drug-target interaction prediction rather than molecule generation (e. g. conducting drug-target binding affinity prediction on more datasets).

3) It is necessary to provide the statistical information  of the dataset.

4) In Figure 2, which part of the model makes the model performance stable under different context affinity?

5) Can OSLS score the topological similarity of molecules? And if so, which part of the experimental results can justify this conclusion?


**Summary Of The Paper:**

The work introduces a novel OSLS (One-Shot Ligand Scoring) model for ligand scoring task to an unseen target based on a single context compound and its experimentally known activity to that target. This model is based on a Siamese-inspired neural network with transformer encoders. The OSLS achieves performance improvements on various benchmarks.

**Summary Of The Review:**

The authors focus on the molecular scoring task, and the model lacks innovation.
The Introduction of the proposed model is clear, but some experiment results do not support its effectiveness for molecule generation.

---

### Decision · Program_Chairs · 2023-01-20

**Decision:**

Reject

**Justification For Why Not Higher Score:**

The 3 issues mentioned above were all significant: lack of relevant baselines, insufficient justification of problem setting, and missing methodological / experimental details.

**Justification For Why Not Lower Score:**

N/A

**Metareview: Summary, Strengths And Weaknesses:**

The paper proposes the One-Shot Ligand Scoring (OSLS) approach to predict the activity of a query compound with an unknown target, based on a single known active drug molecule and its activity level. Their proposed model utilizes two Transformer encoders taking in the query molecule, and the known active drug molecule with its activity level, to predict the activity level of the query molecule.

In general, reviewers agreed that the paper is mostly clear and well-written. However, there are a number of key issues raised by the reviewers, particularly:

- Relevant baselines not being compared: such as (1) MetaDTA, which was mentioned in the experiments section as a baseline but not compared in the main results tables / figures; (2) N-shot methods (the related work mentions that almost all mentioned N-shot methods are for the binary setting, but it is straightforward to adapt them to the regression setting); as well as (3) metric learning approaches. In addition, the molecule generation experiments only use (a differentiable surrogate of) Tanimoto similarity; it is unclear why other neural network approaches would not be applicable here (due to their differentiability).

- Insufficient justification of the 1-shot problem setting: multiple reviewers questioned whether there is sufficient practical justification for the 1-shot setting (as opposed to N-shot, since large parallel screening of compounds would generate multiple hits, and many non-hit compounds). The authors are encouraged to further elaborate on the application scenarios that motivate their setting.

- Experimental / methodology details: such as a clearer description of the linear projection, and dataset statistics / details. Reviewers also suggest the use of other metrics, such as AUPR.

The authors did not post a rebuttal during the review period. In the end, due to the above issues, the work is not ready for publication at ICLR, so I recommend rejection. The reviews offer a number of helpful suggestions for improvement, so I encourage the authors to continue improving the paper based on the reviews.